| 1 | Assessment of meteorological extremes using a synoptic weather generator and a downscaling                                                                 |
|---|------------------------------------------------------------------------------------------------------------------------------------------------------------|
| 2 | model based on analogues                                                                                                                                   |
| 3 | Damien Raynaud <sup>1</sup> , Benoit Hingray <sup>2</sup> , Guillaume Evin <sup>3*</sup> , Anne-Catherine Favre <sup>1</sup> , Jérémy Chardon <sup>1</sup> |
| 4 | 1: Univ. Grenoble Alpes, Grenoble-INP, IGE UMR 5001, Grenoble, F-38000, France                                                                             |
| 5 | 2: Univ. Grenoble Alpes, CNRS, IGE UMR 5001, Grenoble, F-38000, France                                                                                     |
| 6 | 3: Univ. Grenoble Alpes, Irstea, UR ETNA, Grenoble, France.                                                                                                |
| 7 | * <i>Correspondence to</i> : Guillaume Evin (guillaume.evin@irstea.fr)                                                                                     |
| 8 | 1. Abstract                                                                                                                                                |
| 9 | Natural risk studies such as flood risk assessments require long series of weather variables. As an                                                        |

10 alternative to observed series, which have a limited length, these data can be provided by weather 11 generators. Among the large variety of existing ones, resampling methods based on analogues have 12 the advantage of guaranteeing the physical consistency between local weather variables at each time 13 step. However, they cannot generate values of predictands exceeding the range of observed values. 14 Moreover, the length of the simulated series is typically limited to the length of the synoptic 15 meteorological records used to characterize the large-scale atmospheric configuration of the 16 generation day. To overcome these limitations, the stochastic weather generator proposed in this 17 study combines two sampling approaches based on atmospheric analogues: 1) a synoptic weather generator in a first step, which recombines days of the 20<sup>th</sup> century to generate a 1,000-year 18 19 sequence of new atmospheric trajectories and 2) a stochastic downscaling model in a second step, 20 applied to these atmospheric trajectories, in order to simulate long time series of daily regional 21 precipitation and temperature. The method is applied to daily time series of mean areal precipitation 22 and temperature in Switzerland. It is shown that the climatological characteristics of observed 23 precipitation and temperature are adequately reproduced. It also improves the reproduction of 24 extreme precipitation values, overcoming previous limitations of standard analogue-based weather 25 generators. 26

# 27 **2. Introduction**

Increasing the resilience of socio-economic systems to natural hazards and identifying the required 29 adaptations is one of today's challenges. To achieve such a goal, one must have an accurate 30 description of both past and current climate conditions. The climate system is a complex machine 31 which is known to fluctuate at very small time scales but also at large ones over multiple decades or 32 centuries (Beck et al. 2007). It is necessary to study meteorological series as long as possible in order 33 to catch all sources of variability and fully cover the large panel of possible meteorological situations. 34 Regarding weather extremes, the same need arises as estimating return levels associated to large 35 return periods cannot be successfully done without long climatic records (e.g. Moberg et al., 2006; 36 Van den Besserlaar et al., 2013). This comment also applies to all statistical analyses on any derived 37 variable, such as river discharge, for which multiple meteorological drivers come into play and for 38 which extreme events correspond to the combination of very specific and atypical hydro-39 meteorological conditions.

Using weather generators, long simulations of weather variables provide accurate descriptions of the 42 climate system and can be used for natural hazard assessments. Among the large panel of existing 43 weather generators, stochastic ones are used to construct, via a stochastic generation process, single 44 or multisite time series of predictands (e.g. precipitation, temperature) based on the distributional properties of observed data. These characteristics, and consequently the weather generator 45 46 parametrisation, are usually determined on a monthly or seasonal basis to take seasonality into 47 account. They can also be estimated for different families of atmospheric circulation, often referred 48 to as weather types. A state of the art of the most common methods which have been used for the 49 downscaling of precipitation (single or multi-site) is presented in Wilks (2012) or in Maraun et al., 50 (2010). More recent publications gather detailed reviews of some sub-categories of weather 51 generators (e.g. Ailliot et al., 2015 for hierarchical models). An increasing number of studies focuses 52 on the generation of multivariate and/or multi-site series of predictands (e.g. Steinschneider and 53 Brown, 2013; Srivastav and Simonovic, 2015; Evin et al. 2018a; Evin et al. 2018b). Stochastic weather 54 generators are able to produce large ensembles of weather time series presenting a wide diversity of 55 multiscale weather events. For all these reasons, they have been used for a long time to enlighten 56 the sensitivity and possible vulnerabilities of socio-eco-systems to the climate variability (Orlowsky et 57 al. 2010) and to weather extremes.

59 Other models used for the generation of weather sequences are based on the analogue method. 60 Since the description of the concept of analogy by Lorenz (1969), the analogue method has gained 61 popularity over time for climate or weather downscaling. This analogue model strategy has been 62 applied in many studies (Boe et al., 2007; Abatzoglou and Brown, 2012; Steinschneider and Brown 63 2013) and has been used to address a wide range of questions from past hydroclimatic variability 64 (e.g. Kuentz et al, 2015; Caillouet et al., 2016) to future hydrometeorological scenarios (e.g. Lafaysse 65 et al., 2014; Dayon et al., 2015). The standard analogue approach hypothesises that local weather 66 parameters are steered by synoptic meteorology. A set of relevant large scale atmospheric predictors 67 is used to describe synoptic weather conditions. From the atmospheric state vector, characterizing 68 the synoptic weather of the target simulation day, atmospheric analogues of the current simulation 69 day are identified in the available climate archive. Then, the analogue method makes the assumption 70 that similar large scale atmospheric conditions have the same effects on local weather. The local or 71 regional weather configuration of one of the analogue days is then used as a weather scenario for 72 the current simulation day. The key element of the analogue method is that it does not require any 73 assumption on the probability distributions of predictands. This is a noteworthy advantage for 74 predictands, such as precipitation, which have a non-normal distribution with a mass in zero. Most of 75 the studies using analogues focused on precipitation and temperature either for meteorological 76 analysis (Chardon, 2014; Daoud, 2016), or as inputs for hydrological simulations (Marty, 2012; 77 Surmaini et al., 2015). Nevertheless, analogues are increasingly used for other local variables such as 78 wind, humidity (Casanueva et al., 2014) or even more complex indices (e.g. for wild fire, Abatzoglou 79 and Brown, 2012). When multiple variables are to be downscaled simultaneously, another major 80 advantage of the analogue method is that the different predictands scenarios are physically 81 consistent and the simulated weather variables are bound to reproduce the correlations between 82 the variables (e.g. Raynaud et al., 2017) and sites (Chardon et al., 2014). Indeed, when analogue 83 models use the same set of predictors (atmospheric variables and analogy domains) for all 84 predictands, all surface weather variables and sites are sampled simultaneously from the historical 85 records, thus preserving inter-site and inter-variable dependency.

86

The two simulation approaches (stochastic weather generators and analogue methods) described above present some important advantages for the generation of long weather series but also some sizeable drawbacks. Indeed, stochastic weather generators rely on strong assumptions about the statistical distributions of predictands. Identifying the relevant mathematical representations of the

91 processes and achieving a robust estimation of their parameters can be difficult, especially if the 92 length of the meteorological records is short. Modelling the spatial-temporal dependency between 93 variables/sites is often another challenge. Conversely, for the analogue-based approaches, the 94 identification of relevant atmospheric variables providing good prediction skills is not straightforward. The limited length of local weather records is also a critical issue since resampling 95 96 past observations restricts the range of predicted values. In particular, the simulation of unobserved 97 values of predictands is not possible. This can be problematic if one is interested in estimating 98 possible extreme values of the considered variable. Furthermore, the information on synoptic 99 atmospheric conditions required by analogue methods are generally coming from atmospheric 100 reanalyses, which also have a limited temporal coverage (e.g. from the beginning of the 20<sup>th</sup> century for ERA20C, Poli et al., 2013) and from the mid-19<sup>th</sup> century for 20cr (Compo et al. 2011). The length 101 102 of the generated time series is thus typically bounded by the length of the reanalyses.

In this study we propose a weather generator (hereafter SCAMP+) building upon the SCAMP 105 approach presented by Chardon et al. (2018) and making use of reshuffled atmospheric trajectories, 106 following some of the developments by Buishand and Brandsma (2001) and Yiou et al. (2014). The 107 weather scenarios generated by SCAMP being limited by the coverage of the climate reanalyses, the 108 SCAMP+ model extends the pool of possible atmospheric trajectories. Using random transitions 109 between past atmospheric sequences, SCAMP+ generates unobserved atmospheric trajectories, on 110 which the 2-stage SCAMP approach can be applied. By exploring a wide variety of atmospheric 111 trajectories, SCAMP+ introduces some additional large-scale variability which improves the 112 exploration of possible weather sequences. In addition, as done in SCAMP (Chardon et al., 2018), the 113 SCAMP+ approach includes a simple stochastic weather generator which is estimated, for each generation day, from the nearest atmospheric analogues of this day. These two steps (random 114 115 atmospheric trajectories and random daily precipitation/temperature values) improve the 116 reproduction of extreme values, overcoming previous limitations of analogue-based weather 117 generators, usually known to underestimate observed precipitation extremes.

These developments are carried out for the exploration of hydrological extremes (extreme floods) of 120 the Aare River basin in Switzerland (Andres et al. 2019a,b). Meteorological forcings, i.e. temperature 121 and precipitation, are thus simulated to be used as inputs of a hydrological model, for different sub-122 basins of the Aare river basin. Meteorological simulations from SCAMP+ have been used in the Swiss 123 EXAR project<sup>1</sup> and have proven its ability to estimate the discharge values associated to very large 124 return periods on the Aare River. In section 2, we describe in details the test region, the data and 125 three simulation approaches (a classical analogue method, referred to as ANALOGUE, SCAMP and 126 SCAMP+). Section 3 presents the main results on both climatological characteristics and extreme 127 values. Section 4 sums up the main outputs of this study and proposes some further developments 128 and analysis.

# **3. Data and Method**

3.1 Studied region

This study is carried out on the Aare River basin which covers almost half of Switzerland (17,700 km<sup>2</sup>). The topography varies greatly within the basin with, on one hand, high mountains on its southern part (maximum altitude of 4270 m, Finsteraarhorn) and on the other hand, plains on the

<sup>&</sup>lt;sup>1</sup> https://www.wsl.ch/en/projects/exar.html

northern part (minimum altitude of 310 m). These different characteristics coupled with the basin
being located at the crossroads of several climatic European influences give a wide diversity of
possible weather situations across the year.

## 140 3.2 Atmospheric reanalysis and local weather data

The application of the analogue method requires a long archive providing an accurate description of 142 both past synoptic weather patterns and local atmospheric conditions. Indeed, a wide panel of 143 meteorological situations available for resampling is necessary in order to identify the best analogues 144 for the simulation (e.g. Van Den Dool et al., 1994; Horton et al., 2017). In most studies, synoptic 145 situations are provided by atmospheric reanalyses. Here, we use the ERA-20C atmospheric reanalysis 146 (Poli et al., 2013) which provide information on large scale atmospheric patterns on a 6 h basis from 147 1900 to 2010. Data are available at a 1.25° spatial resolution. More specifically, the set of predictors 148 used for the identification of atmospheric analogues is made of the geopotential height at 500 and 149 1000 hPa, the vertical velocities at 600 hPa, large scale precipitation and temperature. The 150 justification of these choices will be given in section 3.3.1.

The local and surface weather parameters of interest are retrieved from 105 weather stations for 152 precipitation and 26 weather stations for temperature, which are spread out homogeneously over 153 our target region, as presented on Figure 1. These data are available at a daily time step from 1930 to 154 2014. They have been spatially aggregated in order to obtain daily time series of mean areal 155 precipitation (MAP) and temperature (MAT) for the Aare region. The three weather generators 156 considered in this study aims at producing scenarios of daily time series of MAP and MAT. In this 157 study, a scenario is defined as a possible realization of the climate system under current climate 158 conditions (i.e. the climate observed for the past few decades). It can be noticed that many 159 applications of analogue-based approaches produce simulations at specific weather stations. 160 However, as shown by Chardon et al. (2016) for France, the prediction skill is significantly improved 161 when the prediction is produced for areal averages, which motivates the generation of MAP and 162 MAT values in this study.

Fig.1: The Aare River basin (red) and locations of the different precipitation (dots) and temperature(triangles) stations.

## 168 3.3 Description of the three models

- This section presents the three different models considered and evaluated in this study.

## 171 3.3.1 ANALOGUE: Classical analogue model

The most basic model evaluated in this study, hereafter referred to as ANALOGUE, relies on a 173 standard 2-level analogue method. For each day of the simulation period (1900 – 2010), analogue 174 days are identified from candidate days. The candidate days, extracted from the archive period, i.e. 175 the period on which both predictors and local observations are available (1930 – 2010), are all days 176 of the archive located within a 61-day calendar window centred on the target day. This calendar filter 177 is expected to account for the possible seasonality of the large scale / small scale downscaling relationship. For instance, candidate days for May 15<sup>th</sup> 2000 are selected within the pool of days 178 ranging from April 15<sup>th</sup> to June 14<sup>th</sup> of each year of the archive. 179

The predictors used for the analogues selection been chosen based on Raynaud et al. (2017). They have been shown to guarantee both inter-variable physical consistency and good predictive skills according to the Continuous Ranked Probability Skill Score (CRPSS), for 4 predictands (precipitation, temperature, solar radiation and wind). In the present work, the predictors considered for each level for the two-level analogy are as follows:

The first level of analogy is based on daily geopotential heights at 1000 hPa and 500 hPa (HGT1000,
 HGT500) as proposed by Horton et al. (2012) and Raynaud et al. (2017). The analogy criterion used
 here is the Teweles–Wobus score (TWS) proposed by Teweles and Wobus (1954). This score has
 been found to lead to higher performances than a more classical Euclidian or Malahanobis distance

(Kendall et al. 1983; Guilbault et Obled, 1998; Wetterhall et al., 2005). It quantifies the similarity 190 between two geopotential fields by comparing their spatial gradients. It allows selecting dates that 191 have the most similar spatial patterns in terms of atmospheric circulation. From September to May, 192 the analogy is based on the geopotential fields on both the current day D and its following day D+1 at 193 12UTC. Thereby, the motions of low-pressure systems and fronts are better described and the 194 prediction skill of the method for precipitation is improved (e.g. Obled et al. 2002; Horton and 195 Brönnimann, 2019). In summer, only the geopotential fields on the current day are used as no similar 196 improvement could be found with a two-day analogy. During this first analogy level, 100 analogues 197 are selected for each day of the target period.

- The second analogy level makes a sub-selection of 30 analogues within the 100 analogues identified 199 in the first analogy level. The analogy score used for the selection is the Root Mean Square Error 200 (RMSE). From September to May, the predictors are the vertical velocities at 600 hPa and the large 201 scale temperature at 2 meters. In summer, the vertical velocities but also other predictors such as 202 the Convective Available Potential Energy (CAPE) led to a rather poor prediction of precipitation due 203 to the coarse resolution of the atmospheric reanalysis, which prevent it from providing an accurate 204 simulation of convective processes. Consequently, large scale precipitation from the reanalysis has 205 been used as a predictor instead, resulting in predictive skills similar to the ones obtained for the rest 206 of the year. The different predictor sets retained for summer and the rest of the year illustrate the 207 differences typically observed between seasons for the main meteorological conditions and 208 processes.

The dimensions and position of the different analogy windows used to compute the analogy measures are presented on Figure 2. They follow the recommendations for the analogy windows optimisation presented in Raynaud et al. (2017) for all predictors.

With this 2-step analogy, 30 scenarios of daily MAP and daily MAT are obtained for each day of the

simulation period (1900-2010). Combined with the Schaake Shuffle method described in section
 3.3.4, the application of the ANALOGUE model leads to 30 scenarios of 110-year time series of daily

MAP and MAT.