# Peer review of "2. Introduction"

_Hydrology and Earth System Sciences, 2019_

## Referee Comment (RC1) · Anonymous Referee #1 · 6 Jan 2020

Review of "Assessment of meteorological extremes using a synoptic weather generator and a downscaling model based on analogs", by Raynaud et al.

This paper presents analyses the performance of three stochastic weather generators based on circulation analogs to simulate daily temperature and precipitation over the Aare river basin (Switzerland). The paper is overall interesting (comparison of three models of increasing complexity) and clearly written. Yet, I think that the experimental set up could be improved and some discussions do not seem to be supported by the results or the figures. Therefore, I feel that there is ample room for improvement of the manuscript to optimize its impact.

Major issues

I do not think that stochastic weather generators (especially those based on analogs) are efficient or even useful to simulate long (i.e. multi-annual) sequences of climate variables, because they cannot take low-frequency variability (due to the ocean or global warming) into account. Instead, they can be very useful to simulate very large ensembles of short sequences in a stationary climate. The manuscript never compares long term variability of model simulations and observations, but focuses on seasonal probability distributions. Therefore, the introduction and interpretation should focus on the challenge of reproducing the probability distribution of climate variables, rather than a centennial reconstruction that is not even analyzed. This would also be more relevant for potential users (as claimed in the abstract and introduction), and would make room for comparisons of probability distributions (past vs. present vs. future).

Does the comparison of seasonal precipitations (Fig. 6) depend on choices of predictors to compute analogues, or even how the seasonality is taken into account?

I am surprised that the discussion of the results is so qualitative: the authors show boxplots or return value plots that yield rather small changes, but never compute actual scores of performance that would quantify the performance of the simulations. Continuous Rank Probability Scores (CRPS) or Tallagrand diagrams (or just quantile plots) would be more useful than a subjective appreciation of Fig. 7. I see no discussion of uncertainties of the results (e.g. with respect to model parameters).

My bet for the strange performance of SCAMP+ to simulate a reasonable range of summer temperatures is that summer temperature follow a distribution that depends on the mean state (e.g. Parey, S., Dacunha-Castelle, D., & Hoang, T. H. (2010). Mean and variance evolutions of the hot and cold temperatures in Europe. Climate dynamics, 34(2-3), 345-359.). Just perturbing with a Gaussian distribution with a fixed variance lowers the variance, with respect to the true temperature variance.

Specific points

My notions of Alpine geography are rather limited. Indications of longitude and latitude in Fig. 1 would be useful.

Using geopotential heights for analogs is certainly a good idea, but the authors should be aware of long term trends (due to temperature increase), which induce biases in analog computations, especially in ERA20C. The authors could consider removing such a trend.

The authors compare (with two different visualizations) 1 day, 3 days, 5 days (Fig. 8) and 92 days (Fig. 7a) precipitation values. What is the cut-off duration for which the three weather generators give similar results (Fig. 7a)? If a generalized Pareto distribution was fitted to precipitation, would the ANALOGUE or SCAMP weather generators be within confidence intervals?

---

## Author Comment (AC1) · 17 Feb 2020

*This paper presents analyses the performance of three stochastic weather generators based on circulation analogs to simulate daily temperature and precipitation over the Aare river basin (Switzerland). The paper is overall interesting (comparison of three models of increasing complexity) and clearly written.*

We thank the reviewer for this positive feedback.

*Yet, I think that the experimental set up could be improved and some discussions do not seem to be supported by the results or the figures. Therefore, I feel that there is ample room for improvement of the manuscript to optimize its impact.*

We appreciate these comments which will help us to improve the manuscript. We hope that the proposed modifications will answer your questions.

*C1: I do not think that stochastic weather generators (especially those based on analogs) are efficient or even useful to simulate long (i.e. multi-annual) sequences of climate variables, because they cannot take low-frequency variability (due to the ocean or global warming) into account. Instead, they can be very useful to simulate very large ensembles of short sequences in a stationary climate. The manuscript never compares long term variability of model simulations and observations, but focuses on seasonal probability distributions. Therefore, the introduction and interpretation should focus on the challenge of reproducing the probability distribution of climate variables, rather than a centennial reconstruction that is not even analyzed. This would also be more relevant for potential users (as claimed in the abstract and introduction), and would make room for comparisons of probability distributions (past vs. present vs. future).*

Thank you for those comments. We partly agree with the reviewer's statement. Yes, the majority of stochastic weather generators (WGEN) are not able to simulate the low – frequency variability of the weather. A number of papers have shown that they cannot simulate a relevant interannual variability of precipitation. The case of WGENs based on analogs is different. By construction, they are conditioned by a sequence of large-scale circulation patterns which presents variability at multiple scales, from daily to interannual and even multi-decadal scales. Thus, WGENs based on analogs are also able to generate long sequences (multiple decades) of weather with relevant multi-annual variability as it derives from the one contained in the large-scale forcing data available for the time period considered for the generation. This is obviously a strength of such WGENs and we will correct our manuscript to better emphasize this point. In the SCAMP+ version, note that the additional step which generates non-observed sequences of large-scale circulation, also generates other realizations of low-frequency variations**.**

In the current manuscript, we only present the mean annual cycle of precipitation and the seasonal probability distributions (for observations and simulations). We agree that this does not allow appreciating the ability of ANALOG and SCAMP generators to produce a centennial reconstruction of regional weather (the reconstruction that can be achieved when the generators are forced by reanalyzes). In addition, the reader cannot appreciate the ability of the models to generate relevant multi-annual variability (for the reconstructed periods for ANALOG and SCAMP, or for long periods obtained from resampled large-scale centennial circulation sequences for SCAMP+). This will be clarified in the revised manuscript. We will add figures that show the time series of observed and generated annual variables (e.g. annual mean, seasonal precipitation) over the last century and over different 100-year resampled large-scale circulation sequences. We will also present how the additional SCAMP+ generation step influences the low -frequency variability of the generated scenarios.

*C2: Does the comparison of seasonal precipitations (Fig. 6) depend on choices of predictors
to compute analogues, or even how the seasonality is taken into account?*

In a preliminary work (not shown in the manuscript), we have considered many different versions of each WGEN based on different sets of predictors respectively. The set of predictors considered in our manuscript has been selected so as to maximize the skill of the WGEN for the prediction of the daily precipitation and temperature observations over the whole simulation period (Chardon et al. 2016, Raynaud et al. 2018). The skill is estimated with the Continuous Ranked Probability Skill Score (CRPSS), a probabilistic evaluation score typically used for the verification of probabilistic or ensemble weather forecasts. This will be clarified in the revised manuscript.

These preliminary analyses showed that the results of the generations depend on the choice of predictors used for the analog selection. This could have been presented in our manuscript. However, results obtained with other predictor sets are not really relevant to consider. The lower skill of these other sets for the prediction of daily variables directly translates to a lower skill for the reproduction of observed seasonal probability distributions. A comment will be added in the revised manuscript.

For the second part of the question, the seasonality is accounted for in different ways:

1. As indicated in Section 3.3.1of the current manuscript, the large-scale predictors are likely to differ from one season to the other. In our work, the first level analogy variables used to identify the candidate analog days are the same but the second level analogy variables differ according to the season.  From September to May they are the vertical velocities at 600 hPa and the large scale temperature at 2 meters. In summer, the vertical velocities but also other predictors such as the Convective Available Potential Energy (CAPE) led to a rather poor prediction of precipitation due to the coarse resolution of the atmospheric reanalysis that prevent it from providing an accurate simulation of convective processes. Consequently, large scale precipitation from the reanalysis has been used instead, resulting in predictive skills similar to the ones obtained for the rest of the year.

2. The large-scale / small scale downscaling relationship is likely to differ from one season to the other. To account for this, the candidate analogs are identified within a 2 months calendar moving window centered on the target day (day of simulation). For instance, when the current simulation day is a 6[th] june, all days between the 6[th] of May and the 6[th] of July of each year are considered as candidate analogs. This calendar constraint for the selection of candidates was not indicated in the manuscript and this point will be added in the revised manuscript.

3. A last calendar constraint is used for the first step SCAMP+ (generation of large scale circulation sequences). This constraint is given at l. 268-270 of the present manuscript version: "To insure that two consecutive days of the generated sequences belong to the appropriate season,  the five 2-day analogue sequences are identified within a +/-15-day moving window centred on the calendar day of the target simulation day".

*C3: I am surprised that the discussion of the results is so qualitative: the authors show
boxplots or return value plots that yield rather small changes, but never compute actual
scores of performance that would quantify the performance of the simulations. Continuous
Rank Probability Scores (CRPS) or Tallagrand diagrams (or just quantile plots)
would be more useful than a subjective appreciation of Fig. 7.*

We could present the CRPSS or Tallagrand diagrams obtained for the ANALOG and the SCAMP models. Both models are indeed expected to reproduce the time variations of observed precipitation. This is not the case for SCAMP+. SCAMP+ produces its own trajectories of large-scale

variables. These trajectories are by construction different from the observed one. As a result, the time series of weather variables generated with SCAMP+ are not expected to fit the observed ones. Note in addition that the main interest of SCAMP+ is that is allows to better explore the diversity of weather configurations and sequences. This is highlighted by the larger range obtained with SCAMP+ for different weather characteristics. A quantitative assessment of SCAMP+ via its ability to reproduce the observed sequence of some variables is thus not really relevant nor interesting.

*C4: I see no discussion of uncertainties of the results (e.g. with respect to model parameters).*

It is true that a discussion on this issue should be incorporated. Among the model parameters that can have an impact on the results, we can stress the importance of:
1. the set of predictors used in the selection of analogs.
2. the number of analogs selected as potential candidates (100 for the first level of analogy and 30 for the second level).
3.the transition probability *p* between large scale trajectory in the first-generation step of SCAMP (*p*=1/7 in the manuscript).

*C5: My bet for the strange performance of SCAMP+ to simulate a reasonable range of summer temperatures is that summer temperature follow a distribution that depends on the mean state (e.g. Parey, S., Dacunha-Castelle, D., & Hoang, T. H. (2010). Mean and variance evolutions of the hot and cold temperatures in Europe. Climate dynamics, 34(2-3), 345-359.). Just perturbing with a Gaussian distribution with a fixed variance lowers the variance, with respect to the true temperature variance.*

Thanks for this comment. As discussed at l. 420-428 of the current manuscript, the limitations of SCAMP+ concerning the generation of hot summers and cold winters are very likely related to the temperature increase experienced over the 20th century, which appears clearly when looking at the hottest summers and the coldest winters. Additional experiments will be performed in order to verify this assumption. In details, we will detrend observed temperatures using a regional linear long-term trend, as done in Evin et al. (2018). We will then redo all the analyses on these detrended temperature observations. We expect this pre-processing to solve this particular issue.

Evin, Guillaume, Anne-Catherine Favre, and Benoit Hingray. 2018. "Stochastic Generators of Multi-Site Daily Temperature: Comparison of Performances in Various Applications." *Theoretical and Applied Climatology*, February, 1–14. https://doi.org/10.1007/s00704-018-2404-x.

*C6: My notions of Alpine geography are rather limited. Indications of longitude and latitude in Fig. 1 would be useful.*

Longitudes and latitudes will be added in Fig. 1.

*C7: Using geopotential heights for analogs is certainly a good idea, but the authors should be aware of long term trends (due to temperature increase), which induce biases in analog computations, especially in ERA20C. The authors could consider removing such a trend.*

This issue is indeed potentially critical. We use geopotential heights for the first analogy level in the analog selection. The Teweles–Wobus score (TWS) proposed by Teweles and Wobus (1954) is used there. This score has been found to lead to higher performances than a more classical Euclidian or Malahanobis distance (Kendall et al. 1983; Guilbault et Obled, 1998; Wetterhall et al., 2005). It quantifies the similarity between two geopotential fields comparing their spatial gradients. It allows

selecting dates that have the most similar spatial patterns in terms of atmospheric circulation at a given (or several) geopotential level(s). As a consequence, it does not compare the absolute values of the geopotential fields between 2 days. We are aware that the mean value of geopotential fields is expected to change with regional warming. The Teweless-Wobus has the great advantage to remove the influence of this long term trend, and should therefore avoid biases for the analog identification. A comment will be added on this point.

Kendall, M., Stuart, A., Ord, J.K., 1983. The Advanced Theory of Statistics. Design and Analysis, and Time-series, vol. 3. Oxford Univ Press, New York. 780 p.

Teweles J, Wobus H. 1954. Verification of prognosis charts. Bulletin of the American Meteorological Society 35: 2599–2617.

Guilbaud S, Obled C. 1998. Prévision quantitative des précipitations journalières par une technique de recherche de journées antérieures analogues: optimisation du critère d'analogie (Daily quantitative precipitation forecast by an analogue technique: optimisation of the analogy criterion). Comptes Rendus de l'Académie des Sciences – Series IIA, Earth and Planetary Science Letters 327: 181–188. doi:10.1016/S1251-8050(98)80006-2.

Wetterhall F, Halldin S, Xu CY. 2005. Statistical precipitation downscaling in central Sweden with the analogue method. Journal of Hydrology 306: 174–190. doi:10.1016/j.jhydrol.2004.09.008.

*C8: The authors compare (with two different visualizations) 1 day, 3 days, 5 days (Fig. 8) and 92 days (Fig. 7a) precipitation values. What is the cut-off duration for which the three weather generators give similar results (Fig. 7a)? If a generalized Pareto distribution was fitted to precipitation, would the ANALOGUE or SCAMP weather generators be within confidence intervals?*

We thank the reviewer for this comment. There are two different aspects. First, in Fig.7, we assess some features of the **climatology**, i.e. the mean of the precipitation amounts at the seasonal scale. We also present the seasonality and precipitation values at the monthly scale in Fig. 6. We could also present the same results at the weekly or daily scale, but it does not present so much interest since it will be similar scaled results (i.e. the monthly mean is equal to the daily mean times the number of days in the month). Second, in Fig. 8, we have a look at the features of **extreme values** for different durations. It does make so much sense to assess annual maxima at the monthly scale (i.e. the maxima of 12 values) since they are not "extremes" in the sense of the extreme value theory (i.e. the maxima of samples of infinite size). We will however consider the interest of including results for 10days maxima which are also of relevance for hydrometeorological extremes in the considered catchment. For the last question, we do not know if the confidence interval obtained from a GPD fitted to precipitation observations would match the intervals obtained from the ANALOGUE and SCAMP simulations, but it can be investigated and discussed.

---

## Referee Comment (RC2) · Anonymous Referee #2 · 20 Mar 2020

This paper compares three different weather generators (WG) based on flow analogues and stochastic weather generators, including a new technique. They analyse the properties of the time series produced by these WG, with a focus on their ability to simulate extremes of precipitation and temperature. I find this paper very interesting and well written. I just have a few minor questions for the authors:

- I am not sure I understand how you link the station data with the reanalyses data. If you use them to calculate daily MAP and MAT of your analogues, how do you calculate daily MAP and MAT for the 1900-1930 period for which you have ERA-20C data but no station data ? Do you only use station data as your observation to which you compare

the WG's simulations ?

-Could you specify the distance you use to calculate analogues ?

-Why do you use HGT1000 rather than SLP ?

-The first time you introduce the term "scenario", I would define what you mean by scenario right away. As you surely know this word has several different meanings in climate science so it can be a bit confusing if you do not define it clearly.

-In the discussion, when you discuss the problem to produce temperature extremes because of climate change, you can expend this to extreme precipitations. There is an observed and projected trend on precipitation extremes related to anthropogenic climate change related to the Clausius-Clapeyron relationship.

I thank the authors for this interesting read, it was a good occupation in covid-19 confinement...

---

## Author Comment (AC2) · 26 Mar 2020

*This paper compares three different weather generators (WG) based on flow analogues and stochastic weather generators, including a new technique. They analyse the properties of the time series produced by these WG, with a focus on their ability to simulate extremes of precipitation and temperature. I find this paper very interesting and well written.*

We thank the reviewer for this very positive feedback.

*I just have a few minor questions for the authors.*

We appreciate these comments and we will modify the manuscript accordingly.

*C1: I am not sure I understand how you link the station data with the reanalyses data. If you use them to calculate daily MAP and MAT of your analogues, how do you calculate daily MAP and MAT for the 1900-1930 period for which you have ERA-20C data but no station data ? Do you only use station data as your observation to which you compare the WG's simulations ?*

For the period 1900-1930, it is true that only atmospheric reanalyses are available, as station data are available from 1930 to 2014. As a consequence, possible analogue candidates are only seeked during the period 1930 to 2014, for which reanalysis and station data are available. For the period 1900-1930, analogues are thus sampled among possible dates in the period 1930-2014. The statistical link between the large-scale atmospheric configuration and the small-scale weather is thus inferred from 84 years and used in a "temporal extrapolation mode" to simulate the weather of a period where we only have the large-scale information. This will be clarified in Section 3.3.1.

As a consequence, also, the results of our simulations are compared to observations that do not correspond to the same period. The simulations cover 110 years and the observations 84 years (for all figures from 4 to 8). This is clarified in the figures.

*C2: Could you specify the distance you use to calculate analogues ?*

Analogues are selected according to the Teweles–Wobus score (TWS) proposed by Teweles and Wobus (1954). This score has been found to lead to higher performances than a more classical Euclidian or Malahanobis distance (Kendall et al. 1983; Guilbault et Obled, 1998; Wetterhall et al., 2005). It quantifies the similarity between two geopotential fields comparing their spatial gradients. It allows selecting dates that have the most similar spatial patterns in terms of atmospheric circulation at a given (or several) geopotential level(s). As a consequence, it does not compare the absolute values of the geopotential fields between 2 days. This paragraph will be included in the revised version of the manuscript.

Kendall, M., Stuart, A., Ord, J.K., 1983. The Advanced Theory of Statistics. Design and Analysis, and Time-series, vol. 3. Oxford Univ Press, New York. 780 p.

Teweles J, Wobus H. 1954. Verification of prognosis charts. Bulletin of the American Meteorological Society 35: 2599–2617.

Guilbaud S, Obled C. 1998. Prévision quantitative des précipitations journalières par une technique de recherche de journées antérieures analogues: optimisation du critère d'analogie (Daily quantitative precipitation forecast by an analogue technique: optimisation of the analogy criterion). Comptes Rendus de l'Académie des Sciences – Series IIA, Earth and Planetary Science Letters 327: 181–188. doi:10.1016/S1251-8050(98)80006-2.

Wetterhall F, Halldin S, Xu CY. 2005. Statistical precipitation downscaling in central Sweden with the analogue method. Journal of Hydrology 306: 174–190. doi:10.1016/j.jhydrol.2004.09.008.

*C3: Why do you use HGT1000 rather than SLP ?*

Raynaud et al. 2017 analysed the predictive skills of numerous atmospheric predictors and more particularly of geopotentials. They selected HGT1000 rather than SLP to compare the sensibility of the results to different geopotential levels (500, 700 or 1000hPa). However, when using the Teweles–Wobus distance SLP and HGT1000 would give very similar results as the positions of highs, of lows and the gradients are quite comparable between 1000hPa and ground level.

Raynaud, D., Hingray, B., Zin, I., Anquetin, S., Debionne, S., & Vautard, R. (2017). Atmospheric analogues for physically consistent scenarios of surface weather in Europe and Maghreb. International Journal of Climatology, 37(4), 2160-2176.

*C4: The first time you introduce the term "scenario", I would define what you mean by scenario right away. As you surely know this word has several different meanings in climate science so it can be a bit confusing if you do not define it clearly.*

We thank the reviewer for this suggestion. In this paper, a scenario indicates a possible realization of the climate for the considered period. Indeed, it differs from climate projections which are also including prescribed projections of socioeconomic global changes. A comment will be added in the revised manuscript.

*C5: In the discussion, when you discuss the problem to produce temperature extremes because of climate change, you can expend this to extreme precipitations. There is an observed and projected trend on precipitation extremes related to anthropogenic climate change related to the Clausius-Clapeyron relationship.*

We thank the reviewer for this comment. Indeed, many evidences have shown that temperature and precipitation extremes are expected to increase in many regions of the world as a result of the global warming.

Concerning the increase of temperature extremes, as discussed at l. 420-428 of the current manuscript, the limitations of SCAMP+ concerning the generation of hot summers and cold winters are very likely related to the temperature increase experienced over the 20th century, which appears clearly when looking at the hottest summers and the coldest winters. Additional experiments will be performed in order to verify this assumption. In details, we will detrend observed temperatures using a regional linear long-term trend, as done in Evin et al. (2018). We will then redo all the analyses on these detrended temperature observations. We expect this pre-processing to solve (at least partly) this particular issue.

Concerning the increase of precipitation extremes, it is difficult to illustrate this aspect with our case study. Figure 1 below shows the evolution of annual maxima of temperature (absolute differences compared to the period 1930-1960 for the 26 stations) and precipitation (relative differences compared to the period 1930-1960 for the 105 stations). While the trend for temperature extremes is clearly observed, the trend for precipitation extremes seems absent. However, as the literature indicates siginifcant trends for precipitation extremes in other regions of the world, as well as expected trends in the future, we agree that this aspect deserves an extended comment in the discussion.

[Figure]

[Figure]

Fig. 1: Evolution of annual maxima of temperature (absolute differences compared to the period 1930-1960 for the 26 stations) and precipitation (relative differences compared to the period 1930-1960 for the 105 stations).

Evin, Guillaume, Anne-Catherine Favre, and Benoit Hingray. 2018. "Stochastic Generators of Multi-Site Daily Temperature: Comparison of Performances in Various Applications." *Theoretical and Applied Climatology*, February, 1–14. https://doi.org/10.1007/s00704-018-2404-x.

---

## Author Response (AR1)

**Response to the reviewer #1**

This paper presents analyses the performance of three stochastic weather generators based on circulation analogs to simulate daily temperature and precipitation over the Aare river basin (Switzerland). The paper is overall interesting (comparison of three models of increasing complexity) and clearly written.

We thank the reviewer for this positive feedback.

Yet, I think that the experimental set up could be improved and some discussions do not seem to be supported by the results or the figures. Therefore, I feel that there is ample room for improvement of the manuscript to optimize its impact.

We appreciate these comments which helped us to improve the manuscript.

C1.1: I do not think that stochastic weather generators (especially those based on analogs) are efficient or even useful to simulate long (i.e. multi-annual) sequences of climate variables, because they cannot take low-frequency variability (due to the ocean or global warming) into account. Instead, they can be very useful to simulate very large ensembles of short sequences in a stationary climate. The manuscript never compares long term variability of model simulations and observations, but focuses on seasonal probability distributions. Therefore, the introduction and interpretation should focus on the challenge of reproducing the probability distribution of climate variables, rather than a centennial reconstruction that is not even analyzed. This would also be more relevant for potential users (as claimed in the abstract and introduction), and would make room for comparisons of probability distributions (past vs. present vs. future).

We thank the reviewer for those comments. We partly agree with the reviewer's statement. Yes, the majority of stochastic weather generators (WGEN) are not able to simulate the low – frequency variability of the weather. A number of papers have shown that they cannot simulate a relevant interannual variability of precipitation. The case of WGENs based on analogs is different. By construction, they are conditioned by a sequence of large-scale circulation patterns which presents variability at multiple scales, from daily to interannual and even multi-decadal scales. Thus, WGENs based on analogs are also able to generate long sequences (multiple decades) of weather with relevant multi-annual variability as it derives from the one contained in the large-scale forcing data available for the time period considered for the generation. This is obviously a strength of such WGENs.

In the revised version of the manuscript, a new subsection 4.3 discusses the multi-annual variability, and includes two new figures (9.a and 9.b which show the time series of observed and generated variables at an annual scale, over the last century. These figures clearly illustrate the ability of ANALOGUE and SCAMP to reconstruct the observed series of annual precipitation and temperature. The succession of dry/wet or cold/warm year is well simulated in both temporality and amplitude. In addition, the positive trend in temperature starting in 1980 is also well reproduced with both models. For SCAMP+, four different 100-year resampled large-scale circulation sequences illustrate how the atmospheric situation generation step influences the low-frequency variability of the generated scenarios.

**C1.2: Does the comparison of seasonal precipitations (Fig. 6) depend on choices of predictors to compute analogues, or even how the seasonality is taken into account?**

In a preliminary work (not shown in the manuscript), we have considered many different versions of each WGEN based on different sets of predictors respectively. The set of predictors considered in our manuscript has been selected so as to maximize the skill of the WGEN for the prediction of the daily precipitation and temperature observations over the whole simulation period (Chardon et al. 2016, Raynaud et al. 2018). The skill is estimated with the Continuous Ranked Probability Skill Score (CRPSS), a probabilistic evaluation score typically used for the verification of probabilistic or ensemble weather forecasts. A comment has been added at I. 180-184.

These preliminary analyses showed that the results of the generations depend on the choice of predictors used for the analog selection. This could have been presented in our manuscript. However, results obtained with other predictor sets are not really relevant to consider. The lower skill of these other sets for the prediction of daily variables directly translates to a lower skill for the reproduction of observed seasonal probability distributions.

For the second part of the question, the seasonality is accounted for in different ways:

1. As indicated in Section 3.3.1 of the manuscript, the large-scale predictors are likely to differ from one season to the other. In our work, the first level analogy variables used to identify the candidate analog days are the same but the second level analogy variables differ according to the season. From September to May they are the vertical velocities at 600 hPa and the large scale temperature at 2 meters. In summer, the vertical velocities but also other predictors such as the Convective Available Potential Energy (CAPE) led to a rather poor prediction of precipitation due to the coarse resolution of the atmospheric reanalysis that prevent it from providing an accurate simulation of convective processes. Consequently, large scale precipitation from the reanalysis has been used instead, resulting in predictive skills similar to the ones obtained for the rest of the year.

2. The large-scale / small scale downscaling relationship is likely to differ from one season to the other. To account for this, the candidate analogs are identified within a 2 months calendar moving window centered on the target day (day of simulation). For instance, when the current simulation day is a 6th June, all days between the 6th of May and the 6th of July of each year are considered as candidate analogs. This calendar constraint for the selection of candidates was not indicated in the manuscript and this point has been added to the revised manuscript (l. 176-179).

3. A last calendar constraint is used for the first step of SCAMP+ (generation of large scale circulation sequences). This constraint is given at I. 285-287 of the present manuscript version: "To insure that two consecutive days of the generated sequences belong to the appropriate season, the five 2-day analogue sequences are identified within a +/-15-day moving window centered on the calendar day of the target simulation day (e.g. all June days if the target day is xxxx-06-15th)."

Different modifications have been made to section 3.3.1 in order to clarify these points.

*C1.3: I am surprised that the discussion of the results is so qualitative: the authors show boxplots or return value plots that yield rather small changes, but never compute actual scores of performance that would quantify the performance of the simulations. Continuous Rank Probability Scores (CRPS) or Tallagrand diagrams (or just quantile plots) would be more useful than a subjective appreciation of Fig. 7.*

We could present the CRPSS or Tallagrand diagrams obtained for the ANALOG and the SCAMP models. Both models are indeed expected to reproduce the time variations of observed precipitation. This is not the case for SCAMP+. SCAMP+ produces its own trajectories of large-scale variables. These trajectories are by construction different from the observed one. As a result, the time series of weather variables generated with SCAMP+ are not expected to fit the observed ones.

Note in addition that the main interest of SCAMP+ is that it allows exploring a greater diversity of weather configurations and sequences. This is highlighted by the larger range obtained with SCAMP+ for different weather characteristics. A quantitative assessment of SCAMP+ via its ability to reproduce the observed sequence of some variables is thus not really relevant nor interesting.

**C1.4: I see no discussion of uncertainties of the results (e.g. with respect to model parameters).**

It is true that a discussion on this issue should be incorporated. Among the model parameters that can have an impact on the results, we can stress the importance of:

1. the set of predictors used in the selection of analogs.

2. the number of analogs selected as potential candidates (100 for the first level of analogy and 30 for the second level).

3. the transition probability p between large scale trajectory in the first-generation step of SCAMP (p=1/7 in the manuscript).

A paragraph has been added at I. 450-455 in the section "Discussion".

C1.5: My bet for the strange performance of SCAMP+ to simulate a reasonable range of summer temperatures is that summer temperature follow a distribution that depends on the mean state (e.g. Parey, S., Dacunha-Castelle, D., & Hoang, T. H. (2010). Mean and variance evolutions of the hot and cold temperatures in Europe. Climate dynamics, 34(2-3), 345-359.). Just perturbing with a Gaussian distribution with a fixed variance lowers the variance, with respect to the true temperature variance.

Thanks for this comment. First, we would like to stress the fact that a Gaussian distribution is used only in both SCAMP and SCAMP+ models. As only SCAMP+ seems unable to generate extremely hot summers (i.e. mean summer temperature above 17°C), it cannot be attributed to this particular feature. It must also be noticed that the Gaussian distribution is applied to the 30 mean areal temperature values obtained from the analogues, for each prediction day. This setting is quite different from Parey, S., Dacunha-Castelle, D., & Hoang, T. H. (2010) who analyze directly temperature series.

Additional analyses have been made to investigate this particular issue for SCAMP+. One hypothesis is that the positive trend in temperature experienced over the 20th century would be responsible for this limitation. Indeed the new weather associations made by the random atmospheric trajectories are mixing days from the 1900s with other from the 2000s, their geopotential analogy being their only selection criteria. This could result in less chance to generate very hot summers (as observed in 2003) or very cold winters (as experienced in 1963). To verify the statement, the boxplot on winter and summer temperature have been regenerated using a modified observed series of temperature for which the trend have been removed (following the method proposed to Evin et al., 2018b, see their section 2.2.1). The results are presented on Fig. R1 below and should be compared to Fig. 7b in the manuscript. One can notice that very little improvement can be seen and that the global temperature increase is not responsible for the non-occurrence and of extremely cold winters and hot summers in SCAMP+ scenarios. Another hypothesis would consist in questioning the predictor used to generate the atmospheric trajectories in SCAMP+. In the approach presented in this study, we used the geopotential height at 1000hPa on two consecutive days. Such a choice guarantees similar positions of high/low pressure systems and comparable movements of these features for the target day and its analogues. However, no conditions on the air mass temperature have been included. In the observed archive of synoptic meteorology, two similar geopotential fields can have rather different air mass temperature. Thus, using only HGT1000 as predictor, only guarantees that the transition from one atmospheric trajectory to another is correct in terms of anticyclonic or unsettled weather. This might results in breaking series of hot/cold days. A possible improvement of

the method would be to explore more complex analogy models to generate the atmospheric trajectories in order to improve the simulation and heat and cold waves. A comment has been added at I. 470-478 to include this discussion.

Fig.10: Observed and simulated boxplots of mean seasonal temperature for models ANALOGUE, SCAMP and SCAMP+. The temperature trend due to climate change has been removed in the daily observed time series which is used thereafter to compute the OBS boxplot and to feed all three models. (Summer: June, July, August. Winter: December, January, February).

Evin, Guillaume, Anne-Catherine Favre, and Benoit Hingray. 2018. "Stochastic Generators of Multi-Site Daily Temperature: Comparison of Performances in Various Applications." *Theoretical and Applied Climatology*, February, 1–14. https://doi.org/10.1007/s00704-018-2404-x.

*C1.6: My* notions of Alpine geography are rather limited. Indications of longitude and latitude in Fig. 1 would be useful.

Longitudes and latitudes have been added in Fig. 1.

C1.7: Using geopotential heights for analogs is certainly a good idea, but the authors should be aware of long term trends (due to temperature increase), which induce biases in analog computations, especially in ERA20C. The authors could consider removing such a trend.

This issue is indeed potentially critical. We use geopotential heights for the first analogy level in the analog selection. The Teweles–Wobus score (TWS) proposed by Teweles and Wobus (1954) is used there. This score has been found to lead to higher performances than a more classical Euclidian or Malahanobis distance (Kendall et al. 1983; Guilbault et Obled, 1998; Wetterhall et al., 2005). It quantifies the similarity between two geopotential fields by comparing their spatial gradients. It allows selecting dates that have the most similar spatial patterns in terms of atmospheric circulation at a given (or several) geopotential level(s). As a consequence, it does not compare the absolute values of the geopotential fields between 2 days. We are aware that the mean value of geopotential fields is expected to change with regional warming. The Teweless-Wobus has the great advantage to remove the influence of this long term trend, and should therefore avoid biases for the analog identification.

A paragraph has been added at I. 186-191 to indicate the use of this score in the description of the ANALOGUE model: "The analogy criterion used here is the Teweles–Wobus score (TWS) proposed by Teweles and Wobus (1954). This score has been found to lead to higher performances than a more

classical Euclidian or Malahanobis distance (Kendall et al. 1983; Guilbault et Obled, 1998; Wetterhall et al., 2005). It quantifies the similarity between two geopotential fields by comparing their spatial gradients. It allows selecting dates that have the most similar spatial patterns in terms of atmospheric circulation."

In addition, a paragraph has been added to the discussion at I. 486-497 concerning potential longterm trends in the predictors: "Trends in observed predictors and predictands, as a result of global warming, could be an additional issue. For instance, the mean elevation of geopotential fields is often expected to increase with mean temperature. Such trends may be detrimental for the simulations, because the analogues identification process would be carried out in a non-homogenous data-set. In the present work for instance, trends in the second analogy level predictors (VV600, P and T) might result, to some extent, in selecting analogues preferentially within the same decade rather than distant ones. This could then reduce the reshuffling potential of the method. This issue is likely to be less critical for the first analogy level of SCAMP and for the generation of atmospheric trajectories in SCAMP+. In this case, analogues are selected according to the Teweles–Wobus score which compares the shapes of geopotential fields and not their absolute values. Quantifying the similarity between these geopotential fields, instead of differences in magnitude, removes the influence of a potential long term trend in this predictor."

Kendall, M., Stuart, A., Ord, J.K., 1983. The Advanced Theory of Statistics. Design and Analysis, and Time-series, vol. 3. Oxford Univ Press, New York. 780 p.

Teweles J, Wobus H. 1954. Verification of prognosis charts. Bulletin of the American Meteorological Society 35: 2599–2617.

Guilbaud S, Obled C. 1998. Prévision quantitative des précipitations journalières par une technique de recherche de journées antérieures analogues: optimisation du critère d'analogie (Daily quantitative precipitation forecast by an analogue technique: optimisation of the analogy criterion). Comptes Rendus de l'Académie des Sciences – Series IIA, Earth and Planetary Science Letters 327: 181–188. doi:10.1016/S1251-8050(98)80006-2.

Wetterhall F, Halldin S, Xu CY. 2005. Statistical precipitation downscaling in central Sweden with the analogue method. Journal of Hydrology 306: 174–190. doi:10.1016/j.jhydrol.2004.09.008.

C1.8: The authors compare (with two different visualizations) 1 day, 3 days, 5 days (Fig. 8) and 92 days (Fig. 7a) precipitation values. What is the cut-off duration for which the three weather generators give similar results (Fig. 7a)? If a generalized Pareto distribution was fitted to precipitation, would the ANALOGUE or SCAMP weather generators be within confidence intervals?

We thank the reviewer for this comment. In Fig.7, we assess some features of the **climatology**, i.e. the distribution of the precipitation amounts at the seasonal scale. We also present the seasonality and precipitation values at the monthly scale in Fig. 6. We could also present the same results at the weekly or daily scale, but it does not present so much interest since it will be very similar scaled results (i.e. the monthly mean is equal to the daily mean times the number of days in the month).

In Fig. 8, we have a look at the features of **extreme values** for different durations. When precipitation values are aggregated at higher temporal scales (e.g. at the monthly scale), annual maxima are not "extremes" in the sense of the extreme value theory (i.e. the maxima of samples of infinite size) and their distribution converges slowly to a Gaussian distribution and the highest intensities are tempered. In Fig. R2 below, we replicate Fig. 8 for higher aggregation durations (7 days, 15 days and 30 days). For 15-day annual maxima, differences between the ANALOGUE and SCAMP are less pronounced. For 30-day annual maxima, SCAMP+ still tends to simulate higher intensities than ANALOGUE and SCAMP models.

Concerning the last question, we show 90% - confidence intervals obtained from a GEV distribution fitted to precipitation observations in Fig. R3 below, along with the intervals obtained from the ANALOGUE and SCAMP simulations. Confidence intervals partly overlap but the confidence intervals from the GEV distribution does not recover entirely the bands obtained from the weather generators.